

# Aerosol trends as a potential driver of regional climate in the central United States: Evidence from observations

Daniel H. Cusworth[1], Loretta J. Mickley[2], Eric M. Leibensperger[3], Michael J. Iacono[4]

[1]Department of Earth and Planetary Sciences, Harvard University, Cambridge, 02138, USA
[2]School of Engineering and Applied Sciences, Harvard University, Cambridge, 02138, USA
[3]Center for Earth and Environmental Science, State University of New York at Plattsburgh, Plattsburgh, 12901, USA
[4]Atmospheric and Environmental Research, Lexington, 02421, USA

*Correspondence to*: Daniel H. Cusworth (dcusworth@fas.harvard.edu)

**Abstract.** *In situ* surface observations show that downward surface solar radiation (SWdn) over the central and southeastern United States (U.S.) has increased by 0.58–1.0 $Wm^{-2}a^{-1}$ over the 2000–2014 timeframe, simultaneously with reductions in U.S. aerosol optical depth (AOD) of 3.3–5.0x$10^{-3}a^{-1}$. Establishing a link between these two trends, however, is challenging due to complex interactions between aerosols, clouds, and radiation. Here we investigate the clear-sky aerosol–radiation effects of decreasing U.S. aerosols on SWdn and other surface variables by applying a one-dimensional radiative transfer to 2000–2014 measurements of AOD at two Surface Radiation Budget Network (SURFRAD) sites in the central and southeastern United States. Observations characterized as "clear–sky" may in fact include the effects of thin cirrus clouds, and we consider these effects by imposing satellite data from the Clouds and Earth's Radiant Energy System (CERES) into the radiative transfer model. The model predicts that 2000-2014 trends in aerosols may have driven clear-sky SWdn trends of +1.35 $Wm^{-2}a^{-1}$ at Goodwin Creek, MS, and +0.93 $Wm^{-2}a^{-1}$ at Bondville, IL. While these results are consistent in sign with observed trends, a cross-validated multivariate regression analysis shows that AOD reproduces 20–26% of the seasonal (June–September, JJAS) variability in clear-sky direct and diffuse SWdn at Bondville, IL, but none of the JJAS variability at Goodwin Creek, MS. Using *in situ* soil and surface flux measurements from the Ameriflux network and Illinois Climate Network (ICN) together with assimilated meteorology from the North American Land Data Assimilation System (NLDAS), we find that sunnier summers tend to coincide with increased surface air temperature and soil moisture deficits in the central U.S. The 1990–2015 trends in the NLDAS SWdn over the central U.S. are also of a similar magnitude as our modeled 2000-2014 clear-sky trends. Taken together, these results suggest that climate and regional hydrology in the central U.S. are sensitive to the recent reductions in aerosol concentrations. Our work has implications for severely polluted regions outside the U.S., where improvements in air quality due to reductions in the aerosol burden could inadvertently increase vulnerability to drought.



# 1 Introduction

From 1930 to 2004, the eastern and central U.S. experienced significant cooling of as much as -0.12 K a$^{-1}$ (Kumar et al., 2013). This phenomenon is known as the "warming hole," as the temperature trend differs in sign from expected greenhouse gas warming (Pan et al., 2004). More recent observations from this region show average annual temperatures increased by

0.6–0.8 K between 1901–1960 and 1991–2012, signaling a reversal of the warming hole trend (Melillo et al., 2014). The causes of the warming hole and its subsequent reversal are uncertain. Previous studies have linked the U.S. warming hole to changing patterns in sea surface temperatures (SSTs) such as the Atlantic Multidecadal Oscillation (AMO) (Kumar et al., 2013; Zhang et al., 2013) or to trends in anthropogenic aerosols, which may influence meteorology by interacting with solar radiation or clouds (Leibensperger et al., 2012b; Booth et al., 2012; Yu et al., 2014). Banerjee et al. (2017) concluded that

while aerosols may have contributed to the warming hole, much of the observed cooling arose from unforced internal variability. Most of these studies relied on global or regional climate models, which are inherently uncertain. In this study, we use recent observations and simple models to better constrain the influence of aerosol–radiation interactions on U.S. regional meteorology. In response to tightening air quality regulations, emissions of aerosol sources are expected to decline worldwide over the 21st century (Westervelt et al., 2015), and so our results could have significance for regional climate

elsewhere.

Some previous model studies have linked the emergence of a warming hole with changes in SST patterns or large-scale circulation. Pan et al. (2004) found that enhanced greenhouse gases produced a circulation response that increased the frequency of the southerly low-level jet over the southern Great Plains in late summer, which in turn replenished soil

moisture and suppressed temperature extremes. They concluded this mechanism could induce a warming hole. Using a climate model forced with observed SSTs, Meehl et al. (2012) linked the warming hole cooling trend to the Interdecadal Pacific Oscillation (IPO). Kumar et al. (2013) found that among 22 climate models, those that had the best representation of the AMO also best reproduced the warming hole, although even these models show large discrepancies with the observed trend.

Alternative explanations for the warming hole involve the influence of aerosol trends on regional meteorology, which may include the impact of changing aerosols on SSTs. Over much of the United States, anthropogenic aerosols are dominated by light-colored, highly reflective species such as sulfate. Such aerosols have adverse health effects, and since the 1970s the U.S. EPA has worked to reduce their sources. Between 1990-2010, the Clean Air Act of 1970 and its amendments cut SO2

emissions by 75 percent (USEPA, 2012), and this reduction may have affected regional climate. Focusing on June-July-August (JJA) during 2000-2011 across the United States, Yu et al. (2014) found positive correlations between monthly mean satellite observations of AOD and cloud optical depth (COD, $r = 0.76$) and between AOD and shortwave cloud forcing (SWCF, $r = 0.84$), as well as negative correlations between SWCF and maximum surface air temperatures ($r = -0.67$). They




thus attribute the 20th century warming hole to aerosol-cloud interactions that lead to surface cooling. To quantify aerosol-radiation interactions, Gan et al. (2014) analyzed a sparse network of surface observations from the Surface Radiation Budget Network (SURFRAD) during 1995–2010, and found increasing trends in annual mean SWdn accompanying decreases in aerosol optical depth (AOD), especially among eastern U.S. sites. However, that study also detected an increase

in clear-sky diffuse radiation, which is perplexing, given that declining aerosol would be expected to decrease such radiation. In contrast, Eshel (2016) diagnosed 1998–2014 surface observations at a site in upstate New York, and inferred that improved air quality has led to a strong increase in JJA SWdn there. The Eshel (2016) result is similar to European studies that have tied aerosol reductions to enhanced SWdn (Philipona et al., 2009; Ruckstuhl et al., 2008).

In a modeling framework, Mickley et al. (2012) found that simple removal of U.S. aerosols exerted a top-of-atmosphere (TOA) radiative forcing of as much as +4-5 $Wm^{-2}$ over the central and eastern U.S. This forcing produced a positive feedback in which increases in surface shortwave radiation (SWdn) enhanced surface fluxes of sensible heat in late summer, drying out soils and reducing cloud cover, which further enhanced SWdn. To understand the climate response of a more realistic representation of U.S. aerosols, Leibensperger et al. (2012a) forced a global climate model using simulated

historical aerosols, and found that high aerosol loading during 1970–1990s increased cloud cover and soil moisture by as much as 5% in the central and eastern U.S. The study also found that aerosol outflow to the Atlantic Ocean in this time frame may have cooled SSTs and increased mean JJA 850 hPa geopotential heights in the region of the Bermuda High (BH), a semi-permanent high-pressure system. Booth et al. (2012) found that stronger aerosol influence on surface forcing in a climate model could better reproduce Atlantic sea surface temperatures (SSTs). Zhang et al. (2013) contested this result,

pointing to mismatches between the Booth et al. (2012) model results and observations of North Atlantic upper ocean heat content and salinity. They proposed instead that variations in the Atlantic Multidecadal Overturning Circulation have driven recent changes in Atlantic SSTs. Mascioli et al. (2016) found competing effects on U.S. temperature extremes by changing aerosols and greenhouse gases over the 20th century, as expected, but temperature in the Southeast responded only weakly to aerosols in their simulation. Finally, using fifty–member ensembles, Banerjee et al. (2017) simulated aerosol-radiation

interactions and the cloud albedo effect on U.S. climate, but not the cloud lifetime effect. They found that aerosol forcing could not entirely explain the 1951–1975 JJA decreasing trend in southeastern U.S. temperatures.

Nearly all these studies on the origin of the U.S. warming hole relied on climate or chemistry–climate models with their many uncertainties. For example, the response of soil moisture or low cloud cover to changing SWdn in such models may

not be well captured (Soden and Held, 2006), and with few observations, aerosol concentrations in the early warming hole years are not well constrained. Aerosol composition is also not well known in the 1950s and 1960s, with black carbon emissions likely uncertain by at least a factor of two (Bond et al, 2007). The meteorological response to black carbon could





be very different to that of sulfate (Koch et al., 2010; Bond et al., 2013), the most abundant anthropogenic aerosol in more recent decades.

In this paper, we turn to observational datasets to try to reconcile the apparently conflicting hypotheses of previous studies
(e.g., Leibensperger et al., 2012b; Kumar et al., 2013; Gan et al., 2014; Yu et al., 2014). We extend previous analyses of SURFRAD trends (Long et al., 2009; Augustine and Dutton, 2013; Gan et al., 2014) by using more recent observations and by focusing on two central and eastern U.S. sites, where emission controls have had the largest influence on AOD. To gain knowledge of the potential influence of changing AOD on regional meteorology, we apply the observed AOD and cirrus cloud variables to a radiative transfer model and a simple statistical model. We further study regional meteorology during
summers with enhanced SWdn to better understand how potential trends in SWdn could influence climate and soil hydrology, especially if the warming hole reversal continues, as suggested by some modeling studies (Leibensperger et al., 2012b). Our work has special relevance for developing countries that currently experience heavy aerosol loading but are planning emission reduction strategies (e.g., Lu et al., 2011).

## 2 Data and Methods

We obtain surface SWdn observations from the SURFRAD Network, which consists of seven sites across the U.S. (Augustine et al., 2000). Although sparse, the network provides some of the longest in situ solar radiance measurements in the U.S., broken into diffuse and direct components. In this study we focus on 2000–2014 data from sites in Bondville, Illinois, and Goodwin Creek, Mississippi, as these sites are located in the central and eastern U.S., and have experienced AOD reductions in the recent past (Gan et al., 2014). We exclude the SURFRAD Penn State site from this study, as the
record is incomplete for much of 2009–2014. SURFRAD solar diffuse radiation is measured through a shaded Eppley Black and White Pyranometer, and direct solar radiation is measured with an Eppley Normal Incidence Pyrheliometer (NIP). Diffuse and direct measurements are summed to produce all-sky shortwave radiation fluxes. All broadband radiation measurements have a three-minute temporal resolution, taken as an average of one-second samples. There are uncertainties of 3% and 6% (4 and 20 Wm$^{-2}$) associated with the direct and diffuse measurements, respectively (Stoffel, 2005), where
uncertainty is derived from the 95% confidence interval. SURFRAD stations also measure AOD in five spectral channels using a multifilter shadowband radiometer. The AOD data are also available as three minute averages, but only under cloud-free conditions. We compare the SURFRAD SWdn radiance data with pyranometer measurements from the U.S. Climate Reference Network (USCRN) (Diamond et al., 2013), but consider only those USCRN sites that have 10+ years of data, starting as early as 2003. We also compare SURFRAD SWdn to in situ pyranometer measurements from the Cary Institute
of Ecosystem Studies (CIES; http://www.caryinstitute.org), located near Millbrook, New York, from 1990–2015.



SURFRAD also provides estimates of total clear-sky radiance using the all-sky observations, following the methods in Long and Ackerman (2000). Briefly, a power law model ($Y = A \times \cos(\theta)b$) is fit, where the initial guess for Y is all-sky SWdn, $\theta$ is the solar zenith angle, and A and b are the fitted coefficients. After eliminating cloudy measurements using various selection criteria, the power law model is refit following an iterative process until its coefficients converge, giving an

estimate of clear-sky fluxes. A weakness of this fitting algorithm is that it may not remove the influence of thin cirrus clouds on the clear-sky flux (Long et al., 2009), and trends in cirrus cloud cover may potentially influence estimates of trends in clear-sky SWdn.

We use the column version of the Rapid Radiative Transfer Model for general circulation models (RRTMG_SW) to relate

SURFRAD radiances to changes in AOD at the two sites (Iacono et al., 2008). RRTMG_SW relies on a correlated-k approach to approximate radiative fluxes and heating rates (Clough et al., 2005); multiple scattering is calculated through a two-stream approximation. We apply monthly mean profiles of atmospheric temperature, pressure, and ozone and water mixing ratios from the MERRA-2 Reanalysis (Rienecker et al., 2011) while keeping all other chemical profiles (e.g., $CO_2$ and $N_2O$) fixed to climatological means. The MERRA-2 ozone product is derived from a simple production and loss

chemical scheme (Suarez et al., 2008), assimilated with measurements from the Ozone Monitoring Instrument and the Microwave Limb Sounder. MERRA-2 ozone fields below 260 hPa are not as reliable. We also apply MERRA-2 surface emissivities to the RRTMG_SW bands in the infrared part of the spectrum (820–4000 cm$^{-1}$), and one minus the observed SURFRAD SW reflectance in the RRTMG_SW bands in the shortwave region (4000–50000 cm$^{-1}$).

We drive the model with observed monthly mean AOD from SURFRAD. For single scattering albedo and asymmetry parameters of the aerosol, we rely on measurements from two long-term AERONET sites located close to the SURFRAD sites: Bondville, IL, and Huntsville, AL (Dubovik and King, 2000). Gan et al. (2014) found close agreement between AOD measurements at the Bondville SURFRAD site and nearby AERONET sites, so we assume that AERONET aerosol properties represent those at SURFRAD. For information on thin cirrus clouds, we rely on cloud fraction, cloud water path,

and ice and liquid radius data retrieved from the CERES instrument onboard the Terra and Aqua satellites (Minnis et al, 2011). CERES thin cirrus cloud optical depths have been shown to correspond those retrieved by the Cloud-Aerosol Lidar and Infrared Pathfinder Satellite (launched in 2006) over land ($r = 0.65$). More detailed information about the cirrus retrieval uncertainty is currently being explored (Minnis, P; personal communication). Since we are interested in total column extinction and surface radiation values, we assume all aerosols are concentrated in the surface layer and fix cirrus fractional

cloudiness at 300 hPa. In a sensitivity simulation, we find that whether we fix the aerosols at the surface layer or distribute the aerosol through the lower troposphere has little effect on modeled surface SWdn. Since the CERES Level 3 retrieval is available only since 2000, we perform all radiative transfer simulations over the 2000-2014 period. Our model setup is similar to the approach of Ruiz-Arias et al. (2013) who performed Weather Research and Forecasting (WRF) model



simulations using RRTMG_SW driven by AERONET AOD and aerosol parameters during October 1–3, 2011. The authors found close agreement between 10-minute modeled and observed total, direct, and diffuse SWdn at SURFRAD and Atmospheric Radiation Measurement (ARM) sites, though they did not perform simulations for Goodwin Creek.

To assess the regional climate impacts of variations in SWdn at Bondville, we use tower data from the nearby Ameriflux site, also in Bondville, and Illinois Climate Network (ICN) sites (WARM, 2014). The Ameriflux site provides 9 years (1998-2007) of continuous observations of summertime radiation, temperature, heat flux, and soil moisture. The tower is located within an active corn/soybean agriculture field, but experiences little irrigation (Meyers, T; personal communication), which could influence the microclimate. The Bondville ICN tower sits in a non-irrigated grass field. Between 1983-2002,
semimonthly soil moisture measurements were made at the tower, using a neutron probe instrument. Since 2003, the station has taken hourly soil moisture measurements using a hydraprobe sensor. The ICN site also provides shortwave global radiation, temperature, and soil temperature data from 1990–present.

We compare Ameriflux and ICN tower data with the estimates from the NASA Land Data Assimilation System (NLDAS)
over North America (Mitchell et al., 2004). The goal of the NLDAS project is to construct high-quality, consistent datasets for use in land surface models (LSMs). NLDAS utilizes a combination of gauge-based precipitation and meteorological data from the NCEP North American Regional Reanalysis (NARR) to drive an ensemble of LSMs, yielding estimates of soil moisture and surface energy fluxes. Version 2 of NLDAS also uses bias-corrected SWdn data from the University of Maryland Surface Radiation Budget dataset, which is based on GOES-8 satellite data (Pinker et al, 2003). Here we analyze
the output from three LSMs in the NLDAS project: Mosaic, Variable Infiltration Capacity (VIC), and Noah (Xia et al, 2012; Koster and Suarez, 1994; Wood et al, 1997). Each LSM has a different treatment of land-atmosphere coupling, but all require that incoming solar radiation balance the sum of outgoing thermal radiation, latent and sensible heat losses, and diffusion of energy into the soil (Overgaard and Rosbjerg, 2011). Spatial resolution of these models is 1/8° x 1/8°.

We calculate annual trends in observations using monthly mean anomalies. For the SURFRAD dataset, we first find the mean diurnal profile for each month during 2000–2014. We then calculate monthly mean SWdn by averaging these diurnal profiles over daylight hours. We compute the monthly climatology over the 2000-2014 period and subtract that from each year's monthly means to arrive at monthly SWdn anomalies. Trends for other data measured with hourly or daily frequency are computed using the same method. Radiative transfer simulations in RRTMG_SW are performed with monthly average
AOD and other environmental variables.  As with the observations, we find monthly anomalies in RRTMG_SW by first calculating the 2000–2014 monthly climatology for each simulation and subtracting that from the corresponding time series of monthly means. We use least squares regression to estimate the slopes of the time series of both observed and modeled monthly anomalies. To test for statistical significance, we follow the method described by Weatherhead et al., (2008), a



method also utilized by Gan et al., (2014). This method determines the significance of a least-squares trend based on variance of the noise (i.e., the residual from the straight line fit), the autocorrelation of the noise, and the number of data points were used to determine the trend. In this study, we set $p \leq 0.05$ as the threshold for statistical significance.

## 3 Long-term trends in surface SWdn

Observed 500 nm AOD decreases significantly at both Bondville (-0.047) and Goodwin Creek (-0.052) during the 2000–2014 time frame (Figure 1), providing evidence of the success of strengthening U.S. air quality regulations. In Figure 2 we show the corresponding trends in observed SWdn for all-sky and clear-sky conditions at the two sites. Both stations show significant increases in total (diffuse + direct) all-sky as well as clear-sky SWdn, as would be expected from the changes in AOD. However, diffuse SWdn is the dominant contributor to these clear-sky trends in both cases, a finding that is discussed further in this section and Section 4.

Figure 2 also shows the modeled trends in SWdn. At Bondville, the aerosol-only simulated trend in clear-sky SWdn agrees in magnitude and sign ($+0.93 \pm 0.22$ $Wm^{-2}a^{-1}$) with that observed ($+0.85 \pm 0.13$ $Wm^{-2}a^{-1}$). Breaking down the total SWdn into its direct and diffuse components, the aerosol-only simulation shows a result consistent with aerosol reductions, specifically large increases in direct SWdn accompanied by a decrease in diffuse SWdn. However, this result differs from clear-sky observations, which show both direct and diffuse SWdn increasing ($+0.41 \pm 0.16$ $Wm^{-2}a^{-1}$ and $+0.44 \pm 0.11$ $Wm^{-2}a^{-1}$).

As noted above, thin cirrus clouds may influence SWdn even under apparently clear-sky conditions. Figure 3 shows spatial trends in cirrus ice cloud fraction over 2000–2014 from CERES. Cirrus cloud fraction increases as much as $+0.5$ % $a^{-1}$ over parts of the eastern U.S., with a $+0.21$ % $a^{-1}$ increase over Bondville. Trends in two other cirrus cloud properties – cloud water path, and cloud particle radius – are not as spatially coherent as those of cirrus cloud fraction, and we do not discuss these further. Incorporating cirrus cloud fraction and the other two cloud parameters into the aerosol-cirrus simulation still yields an increasing trend in total clear-sky SWdn as shown in Figure 2 ($+0.40 \pm 0.29$ $Wm^{-2}a^{-1}$), but with a magnitude only about half that observed. Diffuse SWdn in this simulation is roughly a third of observed clear-sky trend, and this match comes at the expense of direct SWdn, which now shows a decreasing trend, in contradiction to the observations. Neither the direct nor diffuse SWdn trends in the aerosol-cirrus simulation are statistically significant.

At Goodwin Creek, the modeled aerosol-only simulation trend in clear-sky SWdn is $+1.35 \pm 0.25$ $Wm^{-2}a^{-1}$, more than double the observed clear-sky trend ($+0.52 \pm 0.14$ $Wm^{-2}a^{-1}$). As at Bondville, the diffuse component of the observed clear-sky SWdn at Goodwin Creek also exhibits an increasing trend ($+0.34 \pm 0.11$ $Wm^{-2}a^{-1}$), even though cirrus cloud fraction shows no





significant trend there (Figure 3). In fact, the diffuse SWdn trend in the aerosol-cirrus simulation at Goodwin Creek is still negative, though more positive than the aerosol-only simulation. This result contrasts with that in Bondville, where consideration of the large cirrus trend changed the sign and significance of direct and diffuse SWdn trends.

Reconciling the observed trends in diffuse, direct, and total SWdn at the two sites is challenging. In their analysis of SURFRAD data, Gan et al. (2014) also found increasing trends in clear-sky diffuse SWdn averaged over seven sites across the U.S. from 1995–2010. That study hypothesized that trends in this variable could be traced to increasing air traffic and enhanced thin cirrus cloud formation. The effect of aircraft contrails on total cirrus cloud fraction is uncertain. Analyzing trends in upper atmosphere humidity and cirrus cloud cover, Minnis et al. (2004) determined that the observed 1971–1995

+1.0 % per decade (+0.1 % a$^{-1}$) increase in cirrus cloud fraction over the U.S. was indeed caused by increased air traffic. The magnitude of this trend is similar to what we observe in Figure 3. Consistent with Minnis et al. (2004), Travis et al. (2004) found the diurnal temperature range (DTR) over the entire U.S. increased by 1.0 K compared to the 1971–2000 climatological mean after the 3-day suspension of nearly all air traffic following the events on September 11, 2001, with especially large increases (1-2 standard deviations above the climatological mean) in regions such as Illinois that favor the

formation of aircraft contrails at high altitudes. The conclusions of Minnis et al. (2004) finding, although controversial (Hong et al., 2008), suggest a strong influence of contrails on the surface energy budget. At the two SURFRAD sites, we do not find evidence in the diffuse SWdn record of a response to the abrupt halt to air traffic in September 2001 (Figure S1). At Bondville, a slight enhancement in diffuse radiation occurred one day after September 11, which then decayed to the 1995–2000 average after two weeks. Enhancements of similar magnitude occur previous to September 11, so excursions from the

mean may be typical for diffuse SWdn at Bondville. Diffuse SWdn at Goodwin Creek site exhibits a jump on September 11 above the 1995–2000 climatology, but mostly stays within one standard deviation of the 1995–2000 average afterward. These jumps in diffuse SWdn around September 11th would seem to contradict an influence of contrails on surface SWdn observations. However, it is also possible that the sites are simply not representative of the larger domain during this short timeframe.

**4 Short-term variability in SWdn**

The 2000–2014 trends in total observed and modeled clear-sky SWdn are positive at both Bondville and Goodwin Creek, implying a link between aerosols and SWdn. However, surface clear-sky SWdn would be expected to respond rapidly to changes in overhead aerosol, and here we check whether changes in aerosols and/or cirrus clouds can explain the monthly variability of clear-sky SWdn observations. Figure 4 shows the timeseries of observed and standardized monthly mean

SWdn anomalies at Bondville and Goodwin Creek, together with SWdn results from the aerosol-only and aerosol-cirrus radiative transfer simulations. The standardized timeseries is constructed by differencing each month's value with the long-term monthly mean and then dividing by the monthly standard deviation. The Bondville aerosol-only simulations show



greater correlation with observations (e.g., $r = 0.49$ for total SWdn) than that for the aerosol-cirrus simulations ($r = 0.29$). At Goodwin Creek, results from neither monthly simulation are significantly correlated with monthly observations.

To test for a more robust relationship between aerosols and clear-sky SWdn, we develop a statistical model to predict surface

monthly mean clear-sky SWdn anomalies based on AOD and cirrus cloud properties using multivariable linear regression (MLR) with no lag. We perform MLR for June–July–August–September (JJAS). Aerosol load is generally highest in the U.S. during these months (Malm et al., 2004), the incoming solar flux is large, and feedbacks can extend the aerosol influence into late summer (Mickley et al., 2012). We find the optimal coefficients of the MLR by individually fitting independent models across all combinations of predictor variables. For predictor variables, we use AOD and the same cirrus

cloud parameters used to drive RRTMG_SW aerosol-cirrus simulations – cirrus cloud liquid water path (LWP), cirrus cloud ice water path (IWP), cirrus cloud liquid radius ($R_L$), cirrus cloud ice radius ($R_I$), and cirrus cloud fraction ($C_f$). We also include monthly average column ozone (pressure weighted) as a predictor. We optimize coefficients for each individual MLR fit using leave-one-out cross-validation. For each MLR fit, we calculate the Bayesian Information Criterion (BIC), which scores the MLR based on its goodness of fit, and penalizes based on the number of parameters included in the

regression (Posada and Buckley, 2004). Thus we seek solutions that explain clear-sky SWdn using the fewest number of terms, so as to avoid over-fitting. Both clear-sky SWdn and the predictors are detrended, deseasonalized, and standardized before the MLR is fit.

Table 1 summarizes the coefficients of the optimal MLR fits to clear-sky SWdn. At Bondville, we find the optimal clear-sky

total SWdn MLR model is driven by AOD and overhead ozone, explaining 15% of the variance. Direct SWdn is best explained (20%) by AOD alone. The MLR model with AOD and cirrus LWP best fits the observed variability in clear-sky diffuse SWdn, explaining 26% of the variance. The magnitude of the fitted AOD coefficients are of similar magnitude and opposite sign for direct and diffuse SWdn, consistent with the expectation for aerosol-radiation interactions. The coefficients of determination ($R^2$) in Table 1 are similar in magnitude to the correlations between observed SWdn fluxes and those

calculated by the radiative transfer model (Figure 4).

At Goodwin Creek, neither observed AOD nor any property of cirrus clouds can explain the variability in direct or diffuse SWdn, casting doubt on these variables as influences on SWdn at this site. That two separate sites with similar 2000-2014 AOD reductions could have such different MLR results underscores the possible multiplicity of drivers of SWdn and the

uncertainty in resolving local radiation budgets. Though RRTMG_SW driven by AOD has been shown to capture SWdn fluxes at SURFRAD sites well (Ruiz-Arias et al., 2013), averaging daily AOD observations on longer time-scales may bias SWdn radiative transfer calculations, especially in fine aerosol regimes (Ruiz-Arias et al., 2016).





## 5 Meteorological impacts from enhanced SWdn in late summer

Previous work has suggested that the land-atmosphere coupled response to increased SWdn involves a cascade of meteorological phenomena (Shindell et al., 2003; Budyko, 1969). Gu et al. (2012) saw evidence of coupling between net radiation, heat fluxes, and soil moisture using Ameriflux observations during the 2005 growing season in Missouri. The

model study of Mickley et al. (2012) found that U.S. aerosol reductions lead to enhanced latent heat fluxes in early summer which transition to enhanced sensible heat fluxes by late summer/ early fall. We probe the observational record for evidence of these feedbacks by first analyzing 1998–2007 tower data from the Ameriflux site at Bondville. We classify the data for each year into either a sunny or cloudy regime depending on whether the JJAS mean SWdn for that year is above or below the climatological JJAS median. We chose JJAS as the timeframe of reference due to the large summer to early fall SWdn

response from reduced aerosols seen in Mickley et al. (2012). We then compare the responses in monthly mean surface fluxes for these two regimes (Figure 5). We do not consider aerosols directly here, as the short time series of the Ameriflux data limits the ability to assess long-term trends, but we can use our results to understand the regional sensitivity of the land-atmosphere system to SWdn changes driven by AOD trends.

Figure 5 shows that the difference in all-sky daytime SWdn between these two regimes during JJAS is +28.1 Wm$^{-2}$. This difference is about three times the magnitude of the 2000-2014 change in total SWdn observed at Bondville (12.7 Wm$^{-2}$, clear-sky SWdn; +8.7 Wm$^{-2}$, all-sky SWdn). Enhanced SWdn during sunny years leads to increased latent heat fluxes in May-June (+5.3 Wm$^{-2}$), which transition to increased sensible heat fluxes in August–October (+9.5 Wm$^{-2}$). Volumetric soil water content follows the latent and sensible heat fluxes, with greater soil moisture in June (+2.7%) and drier conditions in

August (-3.5%). Because of the increased sensible heating in the sunny regime, we would expect a corresponding change in temperature. However, the difference in JJAS maximum temperature at this site is slightly negative (-0.86 K), with cooler temperatures during sunny years. This negative change in temperature is corroborated by a negative change in upward surface longwave radiation (LWup) at the nearby SURFRAD site (-5.4 Wm$^{-2}$). Precipitation at Bondville does not differ significantly between sunny and cloudy regimes (not shown). Summer 2004, classified as sunny, was paradoxically the

coolest summer in the Ameriflux record, with a mean maximum temperature 1.7 K cooler than the 1998–2007 average, suggesting some other weather phenomenon overwhelming the effects of relatively high incident SWdn.

We next examine ICN measurements of SWdn and surface variables, including soil moisture, using the same method of segregating sunny and cloudy summers. Since the method of measuring ICN soil moisture changed in 2003, we analyze

differences in sunny vs. cloudy regimes within each measurement period, with PER1 corresponding to 1990–2002, and PER2 corresponding to 2003–2014. Though no sensible or latent heat measurements are available here, the ICN data include a record of 4 inch soil temperatures under sod together with soil moisture and maximum air temperatures (Figure 6). We see a similar pattern across the two instrument regime periods, specifically that sunny JJAS summers (PER1 = +16.3 Wm$^{-2}$,





PER2 = +8.5 Wm$^{-2}$) favor hotter soils (PER1 = +1.2 K, PER2 = +2.9 K) and reduced soil volumetric water content (PER1= -3.0%, PER2 = -3.4%). We see that in both instrument periods, the maximum JJAS temperature increases (PER1 = +2.0 K, PER2 = +2.5 K), in contrast with the Ameriflux measurements. In the PER2 ICN data, summer 2004 is categorized as cloudy, hence its cooler temperature is expected. The LWup change at the nearby SURFRAD site during PER2 is consistent

with the temperature increase (+4.8 Wm$^{-2}$). Precipitation again does not differ significantly between sunny and cloudy regimes for each instrument period (not shown). The changes in SWdn for both ICN and Ameriflux stations are strongly positive, with the change during both PER1 and PER2 about half that in the Ameriflux record and about equal in magnitude to the observed change over 2000–2014.

The record of observations at Ameriflux Bondville is relatively short, and the inconsistent soil instrument record at the ICN station complicates long-term analysis. Ameriflux and ICN sites yield consistent SWdn and soil moisture results, but different surface temperature responses, perhaps due to the short-term measurement record. To investigate long-term land-atmosphere changes from 1990–2015, we use assimilated meteorology and other variables from the NLDAS dataset. To assure the quality of the NLDAS driving variables, we compare three of these variables – monthly mean SWdn,

precipitation, and air temperatures – with observations from Ameriflux stations with at least 5 years of data (11 total). We also compare NLDAS SWdn with that from SURFRAD. Figure S2 shows statistically significant agreement among these datasets, with $r$ ranging from 0.60 to 0.77, depending on the variable. We find much weaker, but still statistically significant, agreement between the LSM results and the Ameriflux observations, with $r$ ranging from 0.24 to 0.47, depending this time on both the variable and model (Figure S2). Best matches between models and measurements are obtained for the sensible

heat flux. The relatively weak correlations underline the difficulty in resolving land-atmosphere coupling at 1/8° x 1/8° resolution.

Using the NLDAS dataset and mindful of the uncertainties in LSM results, we expand our focus to look at spatial trends in the relevant variables across the contiguous United States for JJAS over the 1990-2015 time period. Figure 7 shows that

surface JJAS all-sky SWdn has increased significantly by +0.78 Wm$^{-2}$a$^{-1}$ from 1990 to 2015 across the central U.S. (30–50N, 105–85W, denoted by the green box in Figure 7). Figure 7 also reveals a close correspondence between the all-sky NLDAS trend and the all-sky trends derived from site measurements in the SURFRAD (1997–2014), USCRN (2003–2014), and CIES (1990–2015) networks, increasing confidence in the NLDAS dataset. Deviations between data products may be explained by the inconsistent time periods of comparison. Accompanying the change in SWdn is an increase in average JJAS

air temperatures over much of the central and eastern U.S. (0.07 K a$^{-1}$). Precipitation decreases slightly in the central U.S. (-0.19 kg m$^{-2}$ a$^{-1}$), mostly in a few isolated regions over the Great Lakes. The total JJAS enhancement in NLDAS all-sky SWdn over the central U.S. over the 1990-2015 time period is +20 Wm$^{-2}$, similar in magnitude to the increase observed during sunny years at the Ameriflux (+28.1 Wm$^{-2}$) and ICN (PER1= +16.3 Wm$^{-2}$, PER2 = +8.5 Wm$^{-2}$) sites. The 2000–2014





clear-sky SWdn enhancements that we simulate with RRTMG_SW at Bondville and Goodwin Creek (+13.9 and +6.2 Wm$^{-2}$) are about half the NLDAS enhancements averaged over the central U.S. for the longer time period of 1990–2015.

Figure 7 also shows the soil moisture response to increasing SWdn and warmer temperatures over 1990–2015, as calculated by the three LSMs. We combine LSM results by first determining which of the model trends agree in sign in each grid cell and then taking the mean of just those models that agree. The combined trend reveals decreased soil moisture across the central U.S. between 1990 and 2015 (-0.85 kg m$^{-2}$a$^{-1}$, averaged over the region defined in Figure 7), accompanied by an increase in sensible heating in the same region (+0.28 Wm$^{-2}$ a$^{-1}$). This decrease in soil moisture translates to a 1990–2015 decrease in volumetric soil water content of -2.2%, within the range of what is observed as the JJAS difference between sunny and cloudy years at the Ameriflux (-0.14%) and ICN (PER1 = -3.0%, PER2 = -3.4%) sites in Illinois. Since the precipitation patterns seem to be unchanging or small over much of the central U.S. during 1990-2015, the LSM soil moisture results provide evidence of a climate response to greater evapotranspiration in the presence of enhanced SWdn.
We emphasize that our goal in examining the 1990–2015 NLDAS dataset is to probe the regional response of soil moisture and temperature to trends in all-sky SWdn. Although decreasing AOD may have contributed to the all-sky SWdn trends, diagnosing the causes of these trends is beyond the scope of this paper. Because the changes in all-sky SWdn in NLDAS are similar in magnitude to the changes predicted by RRTMG_SW due to the 2000-2014 trend in AOD, we can with greater confidence infer the meteorological effects of the AOD trend with greater confidence.

## 6 Discussion

Here we assemble the evidence of changing surface climate in the U.S. and consider the possible role of aerosols in driving this change. During the 2000–2014 time period, observed AOD decreases significantly at both Bondville, IL (-0.047) and Goodwin Creek, MS (-0.052) Clear-sky total SWdn increases at these sites by 12.7 Wm$^{-2}$ (Bondville) and 7.8Wm$^{-2}$ (Goodwin Creek) over the same time period, suggesting that the declining aerosols are at least partly responsible for these trends. All-sky total SWdn also increases at Goodwin Creek. However, the diffuse component of clear-sky SWdn increases at both Bondville (+6.6 Wm$^{-2}$) and Goodwin Creek (+5.2 Wm$^{-2}$), and the cause of these increases remains an open question.

Using the RRTMG_SW radiative transfer model driven by observed AOD, we simulate increases in total and direct clear-sky SWdn at both sites that are consistent with observations and decreases in diffuse SWdn that are contrary to the observations. Previous studies invoked trends in aircraft contrails to explain the unexpected changes in diffuse SWdn at SURFRAD sites across the U.S. (e.g., Gan et al., 2014), but application of observed cirrus cloud fraction to RRTMG_SW does not resolve this issue. A cross-validated multivariate regression analysis further shows that observed monthly mean AOD accounts for just 20% of the JJAS variability in clear-sky direct SWdn at Bondville, with cirrus cloud liquid water path





and AOD reproducing 26% of the variability in clear-sky diffuse SWdn at this site. No combination of predictors, however, explains the variability of clear-sky direct or diffuse SWdn at Goodwin Creek, casting doubt on the role of aerosol-radiation interactions on local meteorology at this site. Besides AOD and cirrus cloud cover, we are left with few other variables that could influence the direct/diffuse partitioning of clear-sky SWdn, and the cause of the observed increases in SWdn in the

2000-2014 timeframe is not clear.

Our analysis of the Ameriflux data (1998–2007) and ICN data (1990–2014) suggests that soil moisture declines in response to enhanced solar radiation. In particular, we see possible evidence of a soil moisture feedback at the Bondville Ameriflux station, where the difference in JJAS SWdn between sunny and cloudy summers is nearly 30 Wm$^{-2}$, peaking in September.

A sunny summer reduces soil moisture, especially in August (-3.5%), and enhances sensible heat fluxes by +8.7 Wm$^{-2}$, with peak values in September. The SURFRAD data show an all-sky annual SWdn trend of +0.58 Wm$^{-2}$ a$^{-1}$ at Bondville and a +1.0 Wm$^{-2}$ a$^{-1}$ at Goodwin Creek for 2000–2014. This rate translates to changes in SWdn of +8.7 W m$^{-2}$ and +15 W m$^{-2}$ over this time period at these sites, or roughly one-fourth to one-half the change in SWdn between sunny and cloudy years at the Ameriflux station. Despite large spatial heterogeneity, the land surface models in Section 6 show a reduction of volumetric

soil water content of as much as -2.2% over the central U.S. from 1990–2015. Given the observed trends in SWdn, this result is consistent in sign with the 0.14-3.4% decrease in soil moisture content during sunnier summers at the Ameriflux and ICN sites in Illinois. These differences in meteorological variables between the sunny and cloudy regimes are of the same order of magnitude as those simulated by Mickley et al. (2012) for aerosol vs. no aerosol regimes over the eastern US, lending confidence to the conclusions of that model study. Our results are also consistent with Eshel (2016), who found that

the observed SWdn increase from 1988–2014 at a rural site in the Northeast could be explained in a radiative transfer model only when considering trends in anthropogenic aerosols.

Taken together, the observations and modeled results suggest that aerosol-radiation interactions play a small but significant role in regional meteorology at Bondville and the central U.S. Changes in overhead aerosol contribute about one-fourth of

the interannual variability in direct and diffuse clear-sky SWdn and thus the recent decline in AOD may partly account for the observed NLDAS JJAS 1.8 K increase in surface temperatures across the larger region. In contrast, aerosol-radiation interactions do not appear to contribute to the interannual variability in SWdn at Goodwin Creek, casting doubt on the role of these direct interactions in the reversal of the U.S. warming hole in the Southeast. Yu et al., (2014), however, found evidence of aerosol-cloud interactions in the southeastern U.S, so such interactions could potentially be important in that region.


Agriculture is a major industry in the central U.S., and decreases in soil moisture from increased SWdn may have made the region more vulnerable to drought, as suggested by previous model studies (Mickley et al., 2012; Leibensperger et al, 2012b). From our analysis of tower fluxes and the NLDAS assimilation, we find a consistent land-atmosphere response to





SWdn as seen in these model studies. Specifically, soil moisture responds to local enhancements in SWdn that further amplifies SWdn, especially in late summer. Previous studies have diagnosed the strong influence of tropical Pacific SSTs on drought occurrence in the U.S. (e.g., Schubert et al., 2004; Seager and Hoerling, 2014). However, drought models that rely on Pacific SSTs predict a prolonged drought during the 1970s, a period in reality characterized by increased precipitation,

especially in the central and eastern United States (e.g., Seager and Hoerling, 2014). Our work suggests that high loading of anthropogenic aerosol during the 1970s may have led to more moist conditions in the central U.S., countering the SST influence and reducing drought risk. It also underscores the findings of other studies (e.g., Milly and Dunne, 2011) that caution modeling studies against projecting hydrological change in models without finding consistency with surface energy balance changes.

A drawback of this study is that it relies on relatively short-term records of aerosols and surface SWdn. There is also some uncertainty in the SURFRAD measurements, at least when compared to the derived trends per year of SWdn. In trends reported here, however, we find that the standard deviation of the residual noise is greater than the instrument uncertainty. Finally, our study relies on just a few measurement sites to infer relationships of AOD with other variables across a broad

region. The good match between site measurements and assimilated NLDAS data, however, allows us to gain confidence in these inferences.

This study provides observational evidence of the influence of AOD on SWdn and key variables such as soil moisture in the central U.S. By linking trends in AOD, SWdn, and soil moisture, our results point to the importance of considering

atmospheric composition as an additional driver of drought. Currently, many regions of the developing world (e.g., China and India) have much higher aerosol loading than the U.S. and are planning strategies to reduce aerosol sources and improve air quality (Lu et al., 2011). These regions also depend on favorable meteorological conditions for agricultural production to feed growing populations. Our study suggests there may be inadvertent consequences of aerosol reduction on regional climate, including increased risk of drought.

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





**Table 1.** Fitted regression coefficients to downward surface shortwave radiation (SWdn) at Bondville, IL, from 2000–2014. Fits represent the optimal multivariate regression model chosen using the Bayesian Information Criterion (BIC) and fit to column–averaged ozone, aerosol optical depth (AOD), cirrus cloud liquid water path (LWP), cirrus cloud ice water path

5  (IWP), cirrus cloud liquid radius ($R_L$), cirrus cloud ice radius ($R_I$), and cirrus cloud fraction ($C_f$).

| SWdn | Regression Fits[*] | $R^2$ |
|---|---|---|
| Total | -0.27 AOD + 0.30 Ozone | 15% |
| Direct | -0.45 AOD | 20% |
| Diffuse | 0.44 AOD + 0.29 LWP | 26% |

*Predictor variables were detrended, deseasonalized, and normalized by their mean and standard deviation before being fitted to SWdn anomalies.





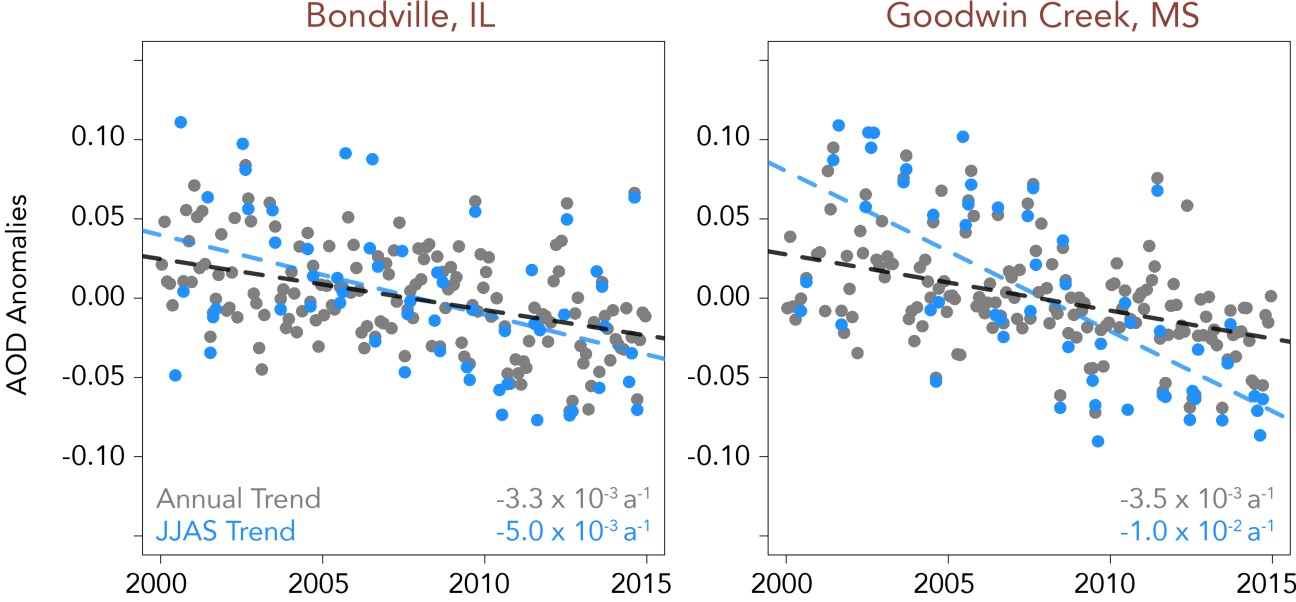

**Figure 1**: Trends in monthly anomalies of 500-nm aerosol optical depth (AOD) at Bondville, IL and Goodwin Creek, MS, during 2000–2014. All trend lines pass the significance threshold (p < 0.05).





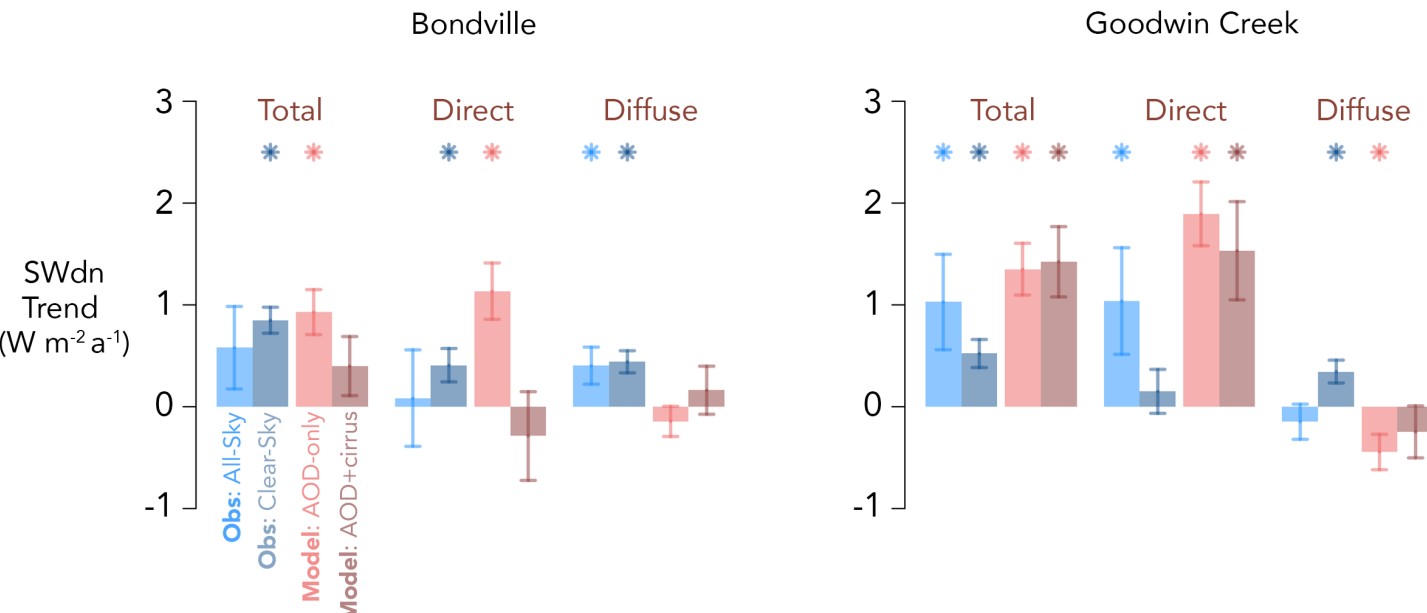

**Figure 2**: Annual trends in downward surface solar radiation (SWdn) during 2000–2014 at Bondville, IL (left), and Goodwin Creek, MS (right). Trends are determined from daytime data (10-23 UTC). Trends in observed all-sky SWdn from the SURFRAD network are shown in light blue; observed trends in clear-sky SWdn are in dark blue. Modeled trends in clear-sky SWdn are shown for an aerosol-only simulation (light red) and for a simulation with both aerosols and cirrus clouds included (dark red). Error bars are the standard error of the slope estimated from a linear regression fit. Asterisks above the bars indicate statistical significance (p < 0.05).




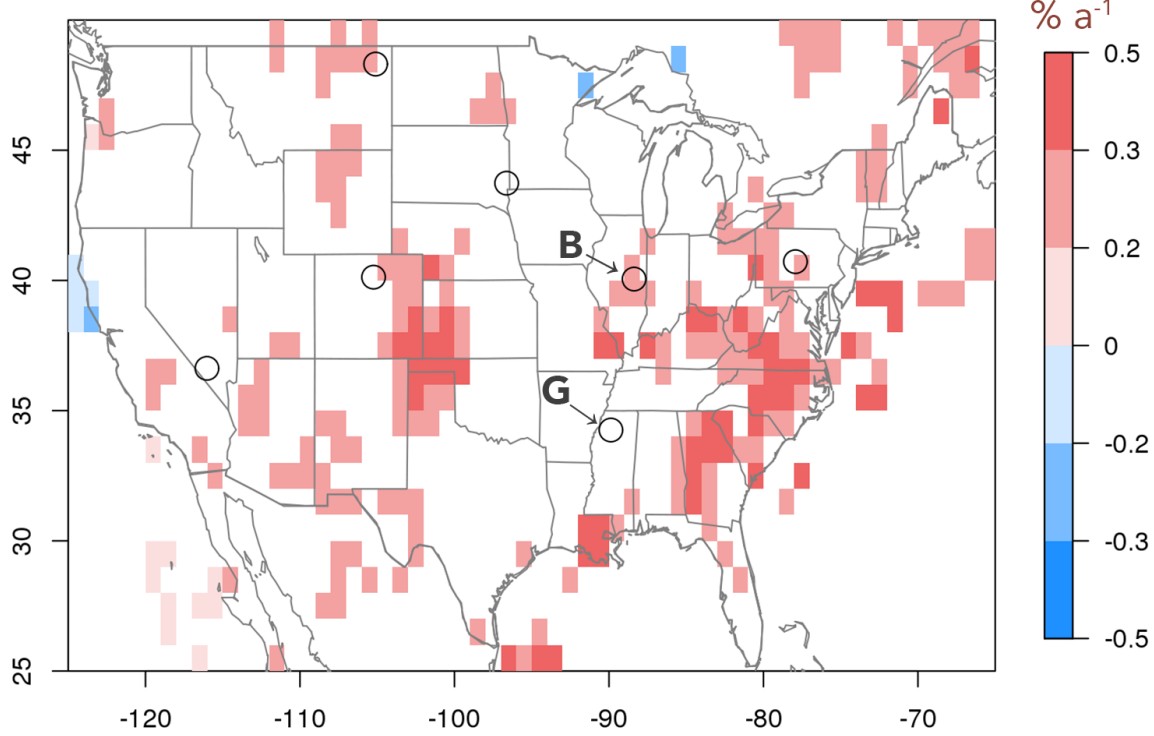

**Figure 3**. Annual trends in cirrus ice cloud fraction as retrieved from Clouds and Earth's Radiant Energy System (CERES) for 2000–2014. Black circles represent the locations of the SURFRAD stations (B = Bondville, IL; G = Goodwin Creek, MS). White indicates regions where trends are not statistically significant ($p > 0.05$).




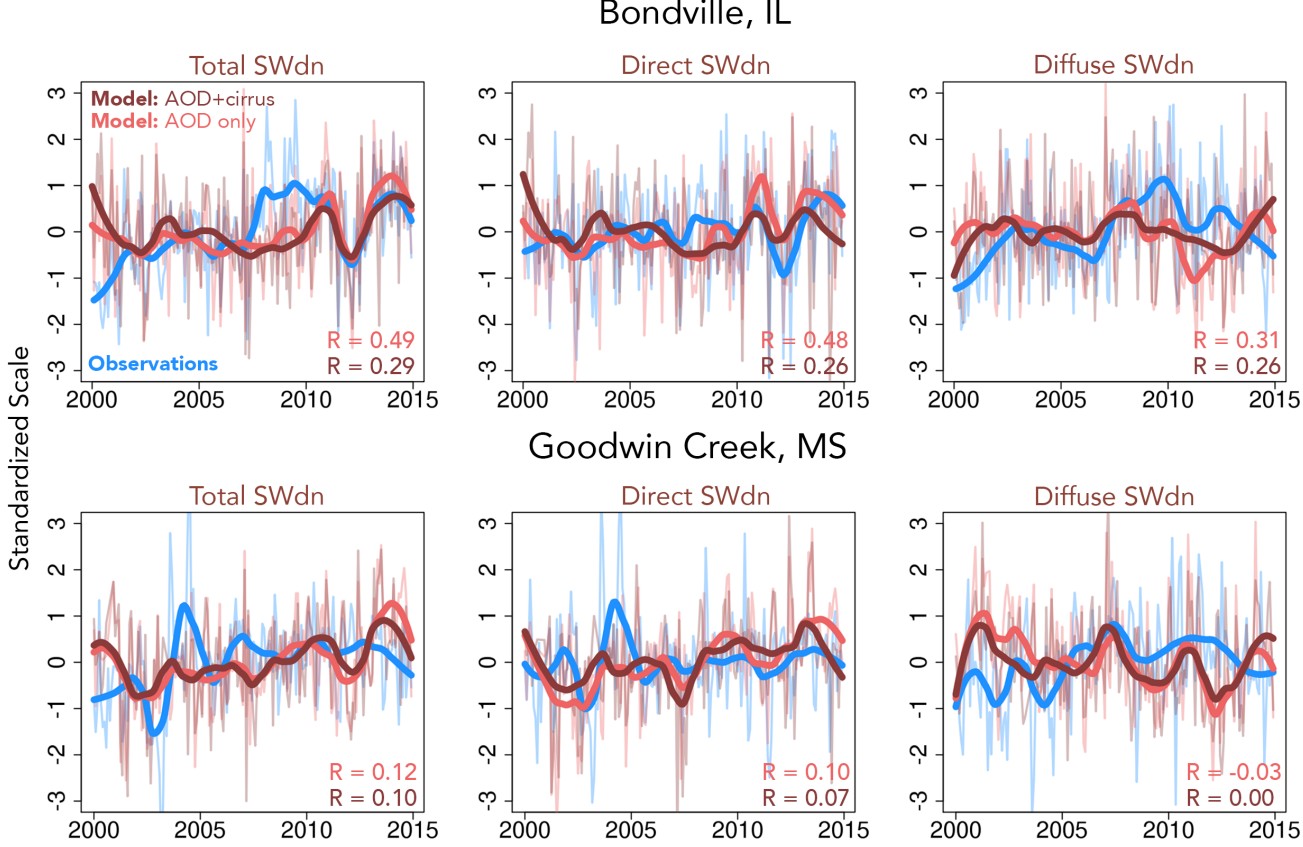

**Figure 4**. Time series of monthly mean anomalies (standardized by mean and standard deviation) in surface solar radiation (shortwave down or SWdn) at Bondville, IL, and Goodwin Creek MS, from 2000–2014. Blue curves denote observations, light red shows model results with observed aerosol optical depths (AOD) taken into account, and dark red shows results when both AOD and observed cirrus cloud fraction are included. Thick lines represent a 3-year locally-weighted scatterplot smoothing (lowess). The correlations $R$ between the non-smoothed model simulations and observations are shown inset. All the correlations at Bondville are significant ($p < 0.05$), and no correlations at Goodwin Creek are significant.



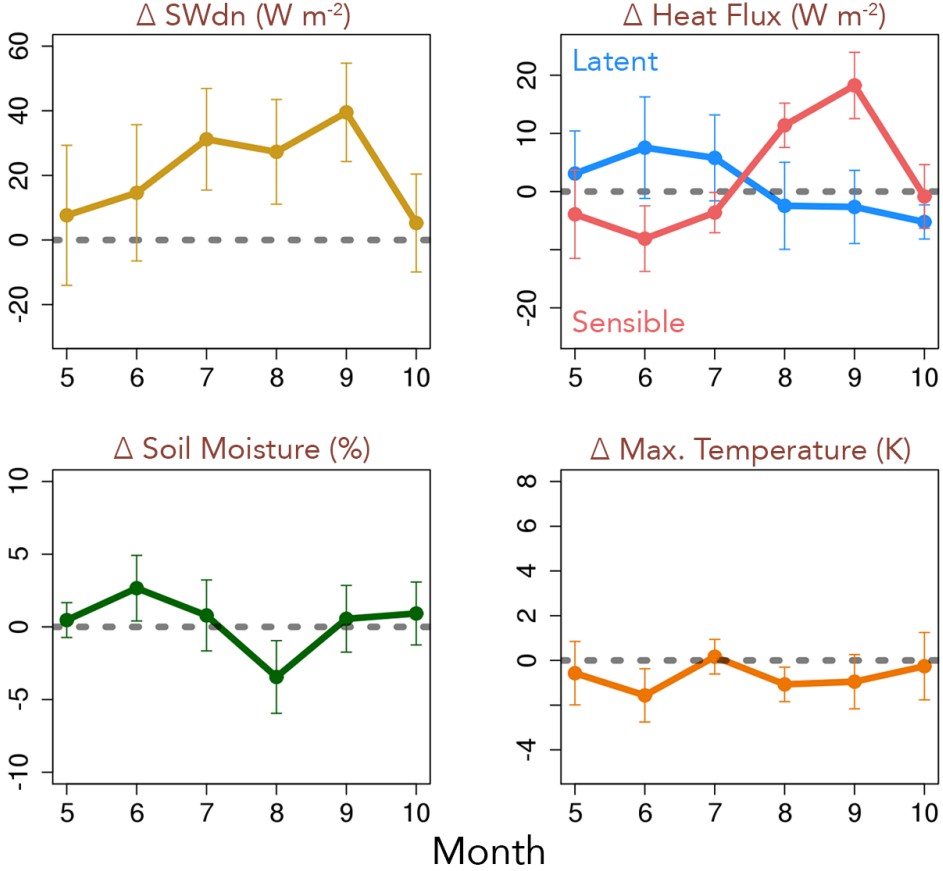

**Figure 5**. Differences in surface measurements at the Bondville Ameriflux site (1998–2007) between sunny and non-sunny summers, where summer is defined as June-July-August-September and a sunny summer is defined as one with average all-sky downward surface solar radiation (SWdn) greater than the median 1998–2007 all-sky summer SWdn. Error bars represent the 95% confidence interval of the difference.





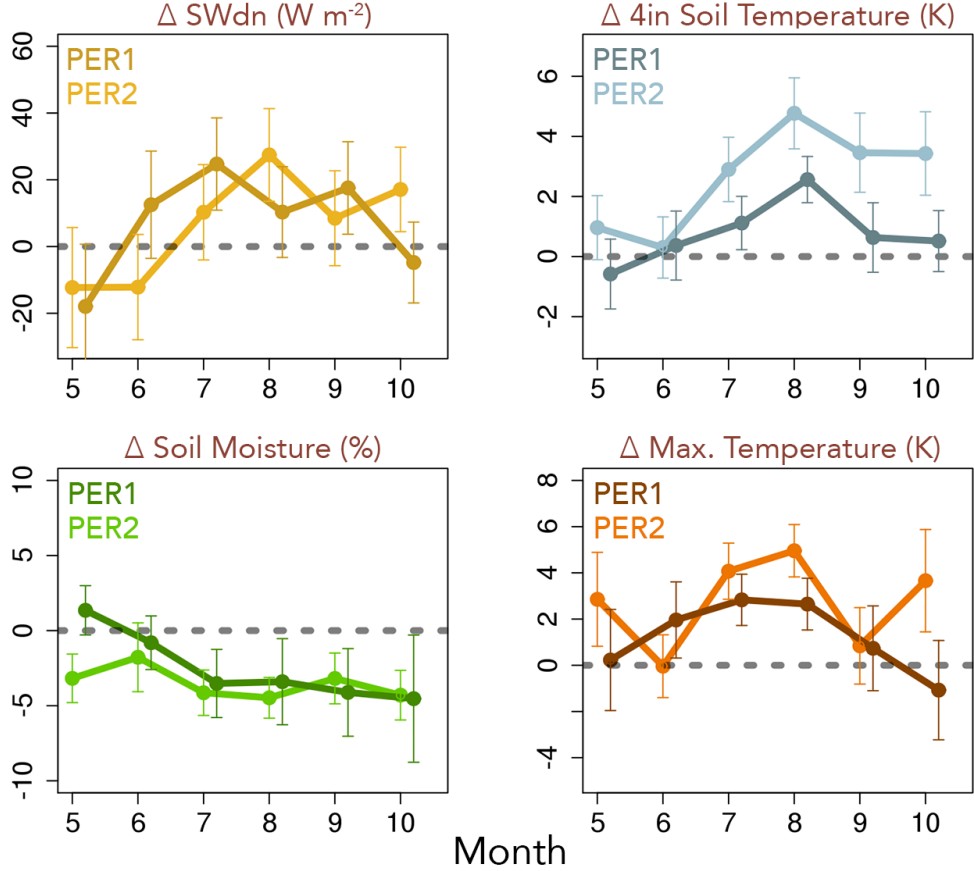

**Figure 6**. Differences in surface measurements at the Bondville ICN site between sunny and non-sunny summers. The technique for measuring soil moisture changed at this site in 2003, and the panels show differences separately for the periods before and after this change: period 1 (PER1, 1990–2002) and period 2 (PER2, 2003–2014). As in Figure 5, summer is defined as June–July–August–September and a sunny summer is defined as one with average all-sky downward surface solar radiation (SWdn) greater than the median all-sky summer SWdn of each respective period. Error bars represent the 95% confidence interval of the difference. For clarity, monthly means for PER1 are slightly offset with respect to the abscissa.




**Figure 7.** June-July-August-September (JJAS) trends in surface meteorological variables from 1990 to 2015. Surface downward solar radiation (SWdn), air temperature, and precipitation are assimilated from observations in the NASA Land Data Assimilation System (NLDAS). Overlaid on the SWdn plot are observed trends from the U.S. Climate Reference Network (USCRN: circles, 2003–2014), SURFRAD (squares, 1997–2015), and the Cary Institute for Ecosystem Studies (CIES: diamond, 1990–2015). The green box in the temperature panel represents the central U.S. (30–50N, 105–85W). In



the soil moisture panel, we combine model results by first determining which of the LSM trends agree in sign in each grid cell and then showing the mean of just those models that agree. White indicates those regions that fail statistical significance (i.e., $p > 0.05$).