# Peer review of "Aerosol trends as a potential driver of regional climate in the central United States: Evidence from observations"

_Atmospheric Chemistry and Physics, 2017_

## Referee Comment (RC1) · Anonymous Referee #2 · 19 Apr 2017

This study explores the connection between changes in regional tropospheric aerosol burden and changes in meteorological parameters measured at two locations in the eastern U.S. – Bondville and vicinity, and Goodwin Creek (with greater emphasis on the former location). The study builds upon previous analyses of the SURFRAD data at these sites by (1) incorporating additional measurements from USCRN, CIES, ICN, Ameriflux, and (2) examining the impact of changes in radiation on surface temperature and soil moisture. As such the results of the analysis are interesting and confirm the complex interactions between aerosols, clouds and radiation. However, in my assessment the usefulness of the manuscript can be improved through additional substantiation of some of the conclusions, and improvements in some aspects of the discussions.

[Figure]

The following suggestions are offered:

1) The goals and objectives of the study should be stated more clearly and supported through the manuscript discussions. The introduction (line 4-5 page 4) suggests that the study goal is to investigate the role of aerosols on the "warming-hole" reversal and to reconcile the conflicting hypothesis on this reversal. One would infer that the driver for this would be the impact of changing aerosol burden on radiation (either through direct or indirect effects). The analysis examines trends in SWdn and its association with trends in aerosols, and attempts to link these with trends in radiation and soil moisture. Yet, the analysis concludes (page 12, line 15) that though decreasing AOD may have contributed to all-sky SWdn trends, diagnosing the causes of the trends is beyond the scope of this paper, which contradicts the objective of reconciling existing conflicting hypothesis was met. There is a lot of good analyses that gets lost in between - perhaps clearly identifying what aspects of the trends, the analysis supports, would help convey a more compelling story.

2) Can Bondville and nearby sites be considered representative of the region where the "warming-hole" has been identified to occur? Some discussion on the suitability of the site for the analysis should be discussed in section 2, especially since the analysis at Goodwin Creek is limited.

3) The analysis attempts to examine the unexpected trends in clear-sky diffuse radiation identified in previous studies (e.g., Gan et al.). While this is a noteworthy attempt, the description of the numerical experiments would benefit from additional details. Specifically, some discussion on how well the CERES retrievals represent the cirrus cloud fraction would be useful to the readers? At locations where the cirrus cloud fractions show strong increasing trends (Figure 3), what trend does the RRTM estimated diffuse radiation show? What may be the likely reasons for the increase in measured diffuse SWdn at Bondville? Also does the CERES cirrus cloud fraction adequately represent aircraft contrails and trends?

4) The RRTMG calculations conducted for the current study should be described in more detail. In particular, what was the temporal resolution of the data input to RRTMG? The discussion on page 6 (lines 29-30) suggest that the AOD and CERES information used in the calculations are monthly means. If finer temporal resolution was used for this data would the results be different? I believe the SURFRAD data is available at finer temporal resolution, but perhaps the CERES is not – some discussion of these aspects of the calculations would be useful in putting the results in context.

5) The purpose (and conclusions) of the MLR analysis in section 4 are not readily apparent. Some discussion on why the MLR captures only 20-26% of the SWdn variability and likely factors influencing the rest would be useful. If increase in clear-sky SWdn (Figure 2) and decrease in AOD (Figure 1) are observed at both Bondville and Goodwin Creek, why does the MLR for Goodwin Creek not capture any of the variability with respect to AOD?

6) Page 9, line 24-25: what is the connection between the $R^2$ in Table 1 and those from the correlation between model and observed SWdn? What does the suggested similarity in these $R^2$ imply?

7) Page 9, line 32: The terminology "fine aerosol regimes" is ambiguous - regimes based on burden, composition, or size?

8) Page 10, lines 16-17: the magnitude of the difference in SWdn between sunny and non-sunny summers is compared with magnitude of 2000-2014 change – It is not apparent to me what significance one can draw from this comparison or what it tells us about the variability in SWdn.

9) Similarly, the significance of the magnitude of change in SWdn between PER1 and PER2, relative to the 2000-2004 change is not obvious?

10) Page 10, line 24: cloudy and relatively cooler conditions across the eastern U.S. for the summer of 2004 are well-documented – it is thus curious why summer 2004 is

classified as sunny?

11) Page 12, line 25: should probably acknowledge that the increase in diffuse component of clear-sky SWdn follows conclusions of earlier studies analyzing SURFRAD measurements.

12) Page 13, line 17-19: This discussion is somewhat vague as it does not state which conclusions of the Mickley et al (2012) study are supported by the current analysis. The sentence should be reworded to clearly state so.

13) Page 13, lines 23-29: It is not clear what to conclude from this discussion which starts by saying that aerosols play a "small but significant" role in regional meteorology (based on Bondville data), then casts doubt on these aerosol-radiation interactions based on analysis of data at Goodwin Creek, but then ends by saying that these interactions could be potentially important. Perhaps aspects of direct and indirect aerosol radiation effects are being mixed and should be explicitly stated. This discussion could be expanded to add clarity.

14) Page 14: line 6: The conclusion that high loading of anthropogenic aerosols during the 1970s contributed to moist conditions, countering the SST influence and reducing drought risk, is somewhat speculative given that no aerosol loading or meteorological parameters from that period are analyzed. Much of the analysis focuses on data from 2000-2014. The extrapolation of results to another period and broader domain should be explained in more detail. In its current form the suggestion is not very convincing.

15) Section 6 would benefit from brief discussion on limitations of the current analysis. While I acknowledge the possible influence of aerosol radiation interactions on precipitation and hydrological cycles, inferences on drought especially in other regions, should be cautiously drawn. If the associations between aerosol burden and radiation are site dependent (as conveyed by the associations at Bondville and Goodwin Creek), what inferences can one draw on the robustness of the association between AOD and soil moisture essentially drawn from data at Bondville (and vicinity sites)? Caveats for

extrapolation of the inferences to other geographic regions should be provided.

16) Page 5, line 26: "correspond those" should be "correspond to those".

---

## Referee Comment (RC2) · Anonymous Referee #3 · 10 Jul 2017

This study 'Aerosol trends as a potential driver of regional climate in the central United States: Evidence from observations' by Cusworth et al., shows links of changes in aerosol burden on surface variables (temperature and soil moisture) and meteorological parameters using In situ observations from two sites over the central and southeastern United States. The problem address in this paper is a relevant scientific question within the scope of ACP. The abstract is concise and and complete. The introduction is set to nice stage and the results are interesting and worth to publish in ACP. I have few minor suggestions:

(a) The statement in abstract "Our work has implications for severely polluted regions

outside the U.S., where improvements in air quality due to reductions in the aerosol burden could inadvertently increase vulnerability to drought". This statement is based on limited variables and I think it is more appropriate if we consider the direct and indirect effect of aerosol. This may be part of another study, but to avoid any misunderstanding, it will be helpful if authors mentioned it clearly in the MS.

(b) The authors used observations from two sites Bondville, Illinois, and Goodwin Creek, Mississippi over the central and eastern U.S. Wonder how these sites are representative of these regions?

(c) On page 6 line 7-10, it is mentioned that tower/sites in Bondville located within an active corn/soybean agriculture field experiences little irrigation which could influence the microclimate. Wonder how much the effect of the irrigation on the observations used in this study, any quantities information will be useful.

(d) Additional details are needed on unit of temperature change as a results of radiative forcing (e.g. what is "a" in the units K aˆ-1 and W mˆ-2 aˆ-1).

---

## Author Comment (AC1) · 29 Jul 2017

We thank the reviewers for their thoughtful comments, which we have addressed below. All page and line numbers refer to those in the revised manuscript.

Since submission of our original manuscript, a new paper focusing on similar issues has been published (Tosca et al., 2017). We have updated the text to mention this paper and compare our results to theirs. See page 3, line 8; page 14, line 23.

Response to Comments from Anonymous Referee #2

1. "The goals and objectives of the study should be stated more clearly and supported

through the manuscript discussions. The introduction (line 4-5 page 4) suggests that the study goal is to investigate the role of aerosols on the "warming-hole" reversal and to reconcile the conflicting hypothesis on this reversal. One would infer that the driver for this would be the impact of changing aerosol burden on radiation (either through direct or indirect effects). The analysis examines trends in SWdn and its association with trends in aerosols, and attempts to link these with trends in radiation and soil moisture. Yet, the analysis concludes (page 12, line 15) that though decreasing AOD may have contributed to all-sky SWdn trends, diagnosing the causes of the trends is beyond the scope of this paper, which contradicts the objective of reconciling existing conflicting hypothesis was met. There is a lot of good analyses that gets lost in between - perhaps clearly identifying what aspects of the trends, the analysis supports, would help convey a more compelling story."

We thank the reviewer for this insight. We now emphasize which conclusions regarding the warming hole hypothesis our analysis of the observations supports.

The radiative transfer/statistical experiments are performed for clear-sky SWdn only, and the purpose of including the clear/sunny analyses are also used to further give evidence that increased SWdn has an observed surface response. We clarify with the following additions.

Page 13, Line 1. "We emphasize that our main goal in examining the 1990–2015 NLDAS dataset is to probe the regional response of soil moisture and temperature to trends in all-sky SWdn. We are not seeking here to understand drivers of SWdn, just the consequences of the variability in SWdn. Coincidentally, the trends in all-sky SWdn trend across the central U.S. in the NLDAS dataset are similar in magnitude to the trends in clear-sky SWdn trends in the RRTMG_SW model at Bondville for a shorter time period. This similarity allows us to infer the potential meteorological consequences of the observed AOD trends with greater confidence.

Page 14, Line 14. "Our Ameriflux, ICN, and NLDAS results show the climate response

to increasing surface SWdn. The RRTMG_SW results in Bondville show that changing aerosols influences SWdn trends."

2. "Can Bondville and nearby sites be considered representative of the region where the "warming-hole" has been identified to occur? Some discussion on the suitability of the site for the analysis should be discussed in section 2, especially since the analysis at Goodwin Creek is limited."

We agree that we want to be clear that the results are site specific. Both the Bondville and Goodwin Creek sites are located in rural regions, away from the city centers.

Page 4, Line 30. "Both SURFRAD sites are located away from urban sources, so we expect them to be representative of the larger region. However, we exercise caution when interpreting site-specific trends."

3. "The analysis attempts to examine the unexpected trends in clear-sky diffuse radiation identified in previous studies (e.g., Gan et al.). While this is a noteworthy attempt, the description of the numerical experiments would benefit from additional details. Specifically, some discussion on how well the CERES retrievals represent the cirrus cloud fraction would be useful to the readers? At locations where the cirrus cloud fractions show strong increasing trends (Figure 3), what trend does the RRTM estimated diffuse radiation show? What may be the likely reasons for the increase in measured diffuse SWdn at Bondville? Also does the CERES cirrus cloud fraction adequately represent aircraft contrails and trends?"

The reviewer asks if CERES retrievals represent cirrus cloud fraction. We point the reviewer to the discussion in the original manuscript.

Page 5, Line 30. "CERES thin cirrus cloud optical depths have been shown to those retrieved by the Cloud-Aerosol Lidar and Infrared Pathfinder Satellite (launched in 2006) over land (r = 0.65)."

The reviewer asks how the RRTMG_SW trends look at sites where CERES cloud fraction is increasing. We show this for Bondville in Figure 2 (and in the discussion starting on page 7, line 26), where RRTMG_SW simulations with AOD+cirrus cloud fraction produce a positive sign in diffuse clear-sky SWdn trends.

Page 7, Line 32. "Diffuse SWdn in this simulation is roughly a third of observed clear-sky trend, and this match comes at the expense of direct SWdn, which now shows a decreasing trend, in contradiction to the observations. Neither the direct nor diffuse SWdn trends in the aerosol-cirrus simulation are statistically significant."

The reviewer asks what could be the cause of the observed clear-sky diffuse SWdn at Bondville. As shown in Figure 2, we could explain the trend using CERES cirrus cloud fraction. However, we draw caution to this conclusion by looking at surface diffuse SWdn during Sept. 11-13, 2001 (starting on page 8, line 11). Since we do not see a significant dip in diffuse SWdn after air traffic was halted, we are cautious in explaining observed diffuse SWdn trends via aircraft contrails, as done by Gan et al. (2014). We add more clarifying language to this section.

Page 8, Line 30. "However, since we do not see strong evidence of aircraft influencing the diffuse clear-sky SWdn at Bondville during Sept. 11-13, we are cautious in ascribing the increasing diffuse clear-sky SWdn trends simulated at Bondville (Figure 2) to aircraft contrails. The increasing trend in CERES cirrus cloud fraction (Figure 3) may be due to other meteorological phenomena that are outside the scope of this paper."

4. "The RRTMG calculations conducted for the current study should be described in more detail. In particular, what was the temporal resolution of the data input to RRTMG? The discussion on page 6 (lines 29-30) suggest that the AOD and CERES information used in the calculations are monthly means. If finer temporal resolution was used for this data would the results be different? I believe the SURFRAD data is available at finer temporal resolution, but perhaps the CERES is not – some discussion of these aspects of the calculations would be useful in putting the results in context."

We perform monthly simulations to bypass inconsistencies with the CERES and AOD

products. We address potential biases on page 10, line 8.

Page 6, Line 3. "We perform monthly simulations to avoid missing data in the AOD and CERES record. Missing daily observations at Goodwin Creek and Bondville range from 43-49%.

Page 10, Line 8. "However, Ruiz-Arias et al. (2016) found a 4% reduction in monthly mean AOD compared to daily AOD in fine aerosol regimes. (Ruiz-Arias et al. segregate aerosol regimes using the Angstrom exponent, which is a direct function of the average size of the aerosol mixture). Among the Bondville and Goodwin Creek AOD timeseries and the CERES retrievals, nearly half the days are missing coincident observations, hence the need to bin the daily data into monthly observations."

5. "The purpose (and conclusions) of the MLR analysis in section 4 are not readily apparent. Some discussion on why the MLR captures only 20-26% of the SWdn variability and likely factors influencing the rest would be useful. If increase in clear-sky SWdn (Figure 2) and decrease in AOD (Figure 1) are observed at both Bondville and Goodwin Creek, why does the MLR for Goodwin Creek not capture any of the variability with respect to AOD?"

The MLR is an effort to show that agreeing AOD and SWdn trends are not coincidental or confounded by other variables. This is now clarified in the text.

Page 9, Line 12. "For AOD to be a controlling factor of clear-sky SWdn, we would expect covariability between variables. To check whether this connection is robust, we develop a statistical model..."

Page 13, Line 27. "With more sites across the Southeast, we could diagnose these inconsistent results as either site-specific, or symptomatic of the larger region."

6. "Page 9, line 24-25: what is the connection between the R2 in Table 1 and those from the correlation between model and observed SWdn? What does the suggested similarity in these R2 imply?"

We now clarify the connection between the correlations in Table 1 and those between the model and observed SWdn.

Page 9, Line 31. "The coefficients of determination ($R^2$) in Table 1 are similar in magnitude to the correlations between observed SWdn fluxes and those calculated by the radiative transfer model (Figure 4), showing a consistency between the two methods to interpret the influence of AOD on SWdn."

7. "Page 9, line 32: The terminology "fine aerosol regimes" is ambiguous – regimes based on burden, composition, or size?"

We now clarify in the text.

Page 10, Line 8, "However, Ruiz-Arias et al. (2016) found a 4% reduction in monthly mean AOD compared to daily AOD in fine aerosol regimes. (Ruiz-Arias et al. segregate aerosol regimes using the Angstrom exponent, which is a direct function of the average size of the aerosol mixture.)

8. "Page 10, lines 16-17: the magnitude of the difference in SWdn between sunny and non-sunny summers is compared with magnitude of 2000-2014 change – It is not apparent to me what significance one can draw from this comparison or what it tells us about the variability in SWdn."

We address the reviewer's concerns in Comment 1.

9. "Similarly, the significance of the magnitude of change in SWdn between PER1 and PER2, relative to the 2000-2004 change is not obvious?"

We address the reviewer's concerns in Comment 1.

10. "Page 10, line 24: cloudy and relatively cooler conditions across the eastern U.S. for the summer of 2004 are well-documented – it is thus curious why summer 2004 is classified as sunny?"

Yes, it is surprising that summer 2004 is classified as sunny in Ameriflux data at

Bondville. We have revised the text.

Page 11, Line 10. "Summer 2004, classified as sunny, was paradoxically the coolest summer in the Ameriflux record, with a mean maximum temperature 1.7 K cooler than the 1998–2007 average at this site. Indeed much of the central U.S. experienced cool temperatures that summer (w2.weather.gov/dtx/2004annualclimatesummary). The surprising result of a cool but sunny summer in 2004 at the Ameriflux site points to the possible problem of relying on its short (10 year) record of observations."

We point the reviewer to the discussion in the original manuscript, where note that in the ICN record (which has more years), 2004 is considered cloudy, as expected.

Page 11, Line 24. In the PER2 ICN data, summer 2004 is categorized as cloudy, hence its cooler temperature is expected.

11. "Page 12, line 25: should probably acknowledge that the increase in diffuse component of clear-sky SWdn follows conclusions of earlier studies analyzing SURFRAD measurements."

Added change to text

Page 13, Line 13: "However, the diffuse component of clear-sky SWdn increases at both Bondville (+6.6 Wm-2) and Goodwin Creek (+5.2 Wm-2), consistent with previous studies (e.g., Gan et al., 2014) and the cause of these increases remains an open question."

12. "Page 13, line 17-19: This discussion is somewhat vague as it does not state which conclusions of the Mickley et al (2012) study are supported by the current analysis. The sentence should be reworded to clearly state so."

This paper supports the conclusion from Mickley et al. (2012) that increased surface SWdn leads to drying of soils, and late summer temperature enhancement over the U.S. Added clarifying language.

Page 14, Line 8: "The observed soil moisture and temperature responses between the sunny and cloudy regimes are of the same order of magnitude as those simulated by Mickley et al. (2012) for aerosol vs. no aerosol regimes over the eastern US, lending confidence to the conclusions of that model study."

13. "Page 13, lines 23-29: It is not clear what to conclude from this discussion which starts by saying that aerosols play a "small but significant" role in regional meteorology (based on Bondville data), then casts doubt on these aerosol-radiation interactions based on analysis of data at Goodwin Creek, but then ends by saying that these interactions could be potentially important. Perhaps aspects of direct and indirect aerosol radiation effects are being mixed and should be explicitly stated. This discussion could be expanded to add clarity."

We are now more careful in our discussion of results at Bondville vs. those at Goodwin Creek. For example, we emphasize that aerosol-radiation interactions are apparent only at Bondville.

Text now reads (Page 14, Line 15): "Taken together, the observations and modeled results suggest that aerosol-radiation interactions play a role in the observed climate trends at Bondville. Changes in overhead aerosol contribute about one-fourth of the interannual variability in direct and diffuse clear-sky SWdn. If the aerosol trends at Bondville are representative of the larger region, the recent decline in AOD may partly account for the 1.8 K increase in surface temperatures across the larger region for JJAS in the NLDAS dataset. In contrast, aerosol-radiation interactions do not appear to contribute to the interannual variability in SWdn at Goodwin Creek, casting doubt on the role of the direct aerosol effect in the reversal of the U.S. warming hole in the Southeast. Yu et al., (2014), however, found evidence of aerosol-cloud interactions in the southeastern U.S., so such interactions could potentially be important in that region.

Our result at Goodwin Creek contrasts with that of Tosca et al. (2017), who concluded

that the observed increase in surface SWdn between 2007 and 2017 at this site was a result of aerosol-radiation interactions. Though we agree with these authors on the sign of the observed surface SWdn trend and find that RRTMG_SW can indeed reproduce a positive SWdn trend when driven by aerosols, we do not find evidence of covariability between AOD and clear-sky SWdn at Goodwin Creek. Hence we disagree that the SURFRAD observations point to aerosol-radiation interactions at this site, as we believe that evidence of covariability between AOD and SWdn is a necessary condition in asserting aerosol-radiation interactions. Yu et al., (2014), however, found evidence of aerosol-cloud interactions in the southeastern U.S., so such interactions could potentially be important in that region."

14. "Page 14: line 6: The conclusion that high loading of anthropogenic aerosols during the 1970s contributed to moist conditions, countering the SST influence and reducing drought risk, is somewhat speculative given that no aerosol loading or meteorological parameters from that period are analyzed. Much of the analysis focuses on data from 2000-2014. The extrapolation of results to another period and broader domain should be explained in more detail. In its current form the suggestion is not very convincing."

We have softened the language

Page 15, Line 6. "We speculate that high loading of anthropogenic aerosol during the 1970s may have led to more moist conditions in the central U.S., countering the SST influence and reducing drought risk. While the model results of Leibensperger et al. (2012a,b) are consistent with this hypothesis, more rigorous model studies with state-of-the-science hydrology and chemistry are needed to confirm it."

15. "Section 6 would benefit from brief discussion on limitations of the current analysis. While I acknowledge the possible influence of aerosol radiation interactions on precipitation and hydrological cycles, inferences on drought especially in other regions, should be cautiously drawn. If the associations between aerosol burden and radiation are site dependent (as conveyed by the associations at Bondville and Goodwin Creek),

what inferences can one draw on the robustness of the association between AOD and soil moisture essentially drawn from data at Bondville (and vicinity sites)? Caveats for extrapolation of the inferences to other geographic regions should be provided."

We address the limitations drawing broad conclusions from site-specific data in the second to last paragraph of the MS.

Page 15, Line 17. "The conflicting results at the Bondville and Goodwin Creek sites demonstrate that caution is needed in extrapolating the aerosol-radiation interactions at Bondville to the entire central and eastern U.S. The good match between site measurements and assimilated NLDAS SWdn data, however, lends confidence that increased surface SWdn has indeed occurred over a broad region. As more in situ measurements of SWdn and AOD are recorded and as USCRN and other national networks are expanded, we expect the discrepancies between SURFRAD sites will be better explained."

16. "Page 5, line 26: "correspond those" should be "correspond to those"."

Fixed.

Response to Comments from Anonymous Referee #3

A. "The statement in abstract "Our work has implications for severely polluted regions outside the U.S., where improvements in air quality due to reductions in the aerosol burden could inadvertently increase vulnerability to drought". This statement is based on limited variables and I think it is more appropriate if we consider the direct and indirect effect of aerosol. This may be part of another study, but to avoid any misunderstanding, it will be helpful if authors mentioned it clearly in the MS."

To make clear we are not looking at drought outside the U.S., we have changed the last sentence of the abstract.

"Our work has implications for severely polluted regions outside the U.S., where improvements in air quality due to reductions in the aerosol burden could inadvertently

pose an enhanced climate risk."

B. "The authors used observations from two sites Bondville, Illinois, and Goodwin Creek, Mississippi over the central and eastern U.S. Wonder how these sites are representative of these regions?"

We agree and thank the reviewer for the comment, as we want to make the distinction that the results are site specific. This comment addressed in Comment 2 from Anonymous Referee #2.

C. "On page 6 line 7-10, it is mentioned that tower/sites in Bondville located within an active corn/soybean agriculture field experiences little irrigation which could influence the microclimate. Wonder how much the effect of the irrigation on the observations used in this study, any quantities information will be useful."

We add clarifying language about the influence of irrigation on the microclimate.

Page 10, Line 26. "We also do not consider microclimate feedbacks from irrigation at the Ameriflux station. Although irrigation has been shown in other studies to suppress extreme temperatures in the Midwest (Mueller et al., 2016), we find that the monthly-average temperatures recorded at the Ameriflux Bondville site correspond closely to the temperature readings at the nearest airport in Champaign, IL (not shown). Also, since the Ameriflux temperatures and SWdn data correspond closely to these value in the 1/8° NLDAS dataset (Figure S2), we assume that the effect of micro-scale irrigation is small."

D. "Additional details are needed on unit of temperature change as a results of radiative forcing (e.g. what is "a" in the units K aЁĘ-1 and W mЁĘ-2 aЁĘ-1).

We follow the SI convention of writing per year as a^-1 (IUPAC Gold Book v2.3.3, pp. 1615). If the paper is accepted, we will follow ACP style rules.

[Figure]

2017.

ACPD

Interactive
comment

---

## Author Response (AR2)

We thank the reviewer for their additional comments to the manuscript, which we have addressed below. All page and line numbers refer to those in the revised manuscript. Reviewer comments are in *italics*, our response is in plain text, and text in the revised manuscript is in blue.

**Response to Comments from Anonymous Referee**

1. *"In the introduction, the authors make the case that given the uncertainties in previous modeling studies and the confounding interpretations of the causes of the "warming hole", they turn to observational data sets to reconcile the conflicting hypotheses of prior studies. The authors have confirmed some of the finding of prior studies (e.g., unexpected trends in diffuse SWdn) but have not necessarily provided new explanations for these. It would be useful to list the key conclusions of this analyses as they pertain to warming hole hypotheses."*

We thank the reviewer for this insight. We now emphasize what specific new findings about warming hole hypotheses we find in this study.

We emphasize that we do not see evidence of aircraft being the driving cause of observed SURFRAD diffuse SWdn.

Page 13, Line 15. "We do not find evidence of aircraft contrails influencing observed diffuse SWdn at SURFRAD sites in the aftermath of Sep. 11th, 2001, and thus caution against attributing increasing clear-sky diffuse SWdn trends to increasing U.S. air traffic as has been done in previous studies (e.g., Long et al., 2009; Augustine and Dutton, 2013)."

We clarify in the discussion where our results conflict with previous studies, and the new insight we provide.

Page 14, Line 18. "Taken together, the observations and modeled results suggest that aerosol-radiation interactions are significant at Bondville, a conclusion doubted in previous studies (Long et al., 2009; Augustine and Dutton, 2013). We find evidence that these interactions play a role in the observed climate trends at Bondville, similar in response to modelling studies that probed the influence of aerosols on the warming hole (Leibensperger et al, 2012b)"

Page 14, Line 28. "Our results underscore the difficulty in attributing the warming hole to only aerosols or only aerosol-radiation interactions using observations. Even though we find evidence of aerosol-radiation interactions at Bondville, our result at Goodwin Creek contrasts with Tosca et al. (2017),"

2. *"In response to my initial review's first comment the authors point to the coincident similar magnitude of all-sky SWdn trend across the central U.S. and clear-sky SWdn trends estimated from RRTMG. How does this similarity allow for greater confidence in inferences of meteorological consequences, as stated in the revised manuscript? Is it reasonable to compare modeled clear sky SWdn trends with observed all sky SWdn trends?"*

We remove confusing language (Page 12, Lline 33) that suggests the scope of this paper is to not assess the drivers all-sky SWdn.

We add language to reaffirm that the aerosol-radiation interactions found in Bondville can be cautiously used as evidence for aerosols influencing all-sky SWdn and meteorology in the region.

Page 13, Line 2. "Given the similar magnitude of the observed all-sky and clear-sky SWdn trends and the covariability of AOD with clear-sky SWdn at Bondville, we infer the potential meteorological consequences of the observed AOD trends in the central U.S. with greater confidence. This inference is limited due to the availability of just one surface site that measures direct and diffuse components of all-sky SWdn, clear-sky SWdn, and AOD, but can be used as an example of assessing aerosol-climate interactions in future studies."

3. *"The authors further point to the above clarification to address the clarification sought in comments 8 and 9 of the previous review, stated again below:*

*- "the magnitude of the difference in SWdn between sunny and non-sunny summers is compared with magnitude of 2000-2014 change – It is not apparent to me what significance one can draw from this comparison or what it tells us about the variability in SWdn"*
*- "Similarly, the significance of the magnitude of change in SWdn between PER1 and PER2, relative to the 2000-2004 change is not obvious?"*

*I do not understand how either of these are clarified?"*

We remove the sentences that compare the Ameriflux and ICN sunny vs. cloudy change in SWdn (Page 11, Line 2; Page 11, Line 27). Now the discussion of trends in SWdn and other climate variables is limited to the section on NLDAS (starting on Page 12, Line 9).

4. *"Page 4, line 30-31: the authors state that they exercise caution in interpreting site-specific trends. The statement alone is a bit vague and does not necessarily support the broader generalizations subsequent conclusions convey (e.g., Godwin Creek analysis that is suggested to "casting doubt on role of direct aerosol effects on reversal of warming hole" across the ebtire Southeast."*

We thank the reviewer for this comment and modify the language used. Specifically, we remove the word "caution" as our goal is to emphasize that site-specific trends should be corroborated by regional trends in order to make broad inferences.

Page 4, Line 31. "To verify consistency of SURFRAD trends against trends across the broader region, we compare the SURFRAD SWdn radiance data with pyranometer measurements ..."

5. *"Later in the conclusion section it is stated that the Bondville and Goodwin Creek results demonstrate that caution should be exercised in extrapolating the results to the entire central and southeastern U.S. It is important to maintain consistent messaging through the discussions and acknowledge that many of the conflicting hypotheses still remain. Clearly stating the (site) specific conclusions and any similarities across them would more clearly portray the analyses and the remaining questions. As such it is not readily apparent which aspects of the conflicting hypotheses this analysis has conclusively addressed."*

We reword the language in the conclusion to emphasize that site-specific trends can be used to extrapolate to the broader region if corroborated by additional data.

Page 15, Line 24. "We find evidence of aerosol-radiation interactions at Bondville but not Goodwin Creek. Though we use site-specific data to come to these conclusions, the good match between site measurements and assimilated NLDAS SWdn data, however, lends confidence that increased surface SWdn has indeed occurred over a broad region."

6. *"Pg 6, line 8-9: I am assuming that monthly simulations imply that monthly mean information was used to drive the RRTMG – please clarify what "monthly simulations" imply."*

We now clarify that we perform simulations with monthly mean information

Page 6, Line 8. "We perform simulations with monthly mean data to avoid gaps in the AOD and CERES record."

7. *"Page 8, line 31: "other meteorological phenomena" are attributed as likely causes for increasing trends in CERES cirrus cloud fraction to rule out possible role of contrails. Some suggestion (qualified as speculative) on these other phenomena would be useful for readers who may want to pursue this further."*

We hesitate to speculate on other specific drivers of cirrus cloud trends. We replace the words "meteorological phenomena" (Page 9, Line 2) with "factors" to keep the array of potential cirrus drivers intentionally wide.

8. *"P 14, line 23-30: I do not fully understand the rationale of this discussion. If RRTMG can reproduce the observed SWdn trend when driven by (observed) aerosol, it implicitly implies co-variability between the aerosol loading (and thus AOD) and SWdn. The discussion is confusing.*

The implicit co-variability the reviewer mentions at Goodwin Creek between AOD and observed SWdn is induced by the trends in both variables. However, the residuals from the respective trends in AOD and observed SWdn do not co-vary at Goodwin Creek (discussed in Section 4). For evidence of aerosol-radiation interactions, we require RRTMG to reproduce the observed SWdn trend, but we also expect the AOD and observed SWdn residuals to co-vary on interannual timescales as well. Tosca et al. (2017) only meets one of these criteria at Goodwin Creek, so we contest their conclusion (starting on Page 14, Line 32).

[revised manuscript text omitted]